# Formulation and *in-vitro* functional evaluation of a Bacillus-based multi-strain probiotic consortium relevant to protein-energy malnutrition

Priya Mori[1], Ishita Modasiya[1], Mehul Chauhan[1], Hina Maniya[1,2], Vijay Kumar[1,3]*, Apurba Kumar Sarkar[4]*

1 Postbiotics and Foodomics Lab, Department of Microbiology, School of Science, RK University, Rajkot, Gujarat, India, 2 Department of Microbiology, Krishna School of Science, Dr. Kiran and Pallavi Patel Global University (KPGU), Vadodara, Gujarat, India, 3 Department of Microbiology and MLT, Bhagwan Mahavir College of Basic and Applied Sciences, Bhagwan Mahavir University, Surat, Gujarat, India, 4 Paradise Scientific Company Ltd., Nazrul Islam Sharak, Dhaka, Bangladesh

* vijaykumar2051983@gmail.com (VK); apurbas@ymail.com (AKS)

## Abstract

Protein-energy malnutrition (PEM) remains a critical global health challenge, characterized by impaired nutrient absorption and chronic gut inflammation. While probiotics offer a potential therapeutic avenue, the efficacy of single-strain interventions is often limited. This study aimed to formulate and evaluate a *Bacillus*-based multi-strain probiotic consortium (MSPC) specifically tailored for PEM. Three strains—*Bacillus spizizenii* BAB 7915, *Bacillus tequilensis*, and *Bacillus rugosus*-were selected based on their non-antagonistic, synergistic growth profiles. The MSPC demonstrated superior functional attributes compared to individual strains, exhibiting significant proteolytic activity ($0.52 \pm 0.03$ U/mL) and robust anti-inflammatory potential ($5.33 \pm 0.06$ U/mL). Additionally, the consortium showed high tolerance to gastrointestinal conditions and strong antioxidant properties. These results suggest that the MSPC can effectively enhance protein hydrolysis and mitigate gut inflammation, providing a scientifically validated, low-cost formulation for the nutritional rehabilitation of PEM patients.

## Introduction

Malnutrition is a significant global health challenge, especially among children. It is broadly categorized into undernutrition and overnutrition. Undernutrition, caused by deficiencies in essential micronutrients and macronutrients, manifests in conditions such as stunting, wasting, and being underweight, while overnutrition primarily leads to obesity. Globally, children under the age of five are particularly vulnerable to undernutrition [1]. India bears one of the highest global burdens of malnutrition, with recent estimates indicating that approximately 35.5% of children under five years of age are stunted, 19.3% are wasted, and 32.1% are underweight (NFHS-5, 2019–2021 [2];

**Data availability statement:** All relevant data are within the manuscript and its Supporting information files.

**Funding:** The author(s) received no specific funding for this work.

**Competing interests:** The authors have declared that no competing interests exist.

UNICEF [3]). Protein-energy malnutrition remains a major contributor to childhood morbidity and mortality, particularly in socioeconomically vulnerable populations. In India, protein-energy malnutrition (PEM) poses a substantial health issue [4]. According to a report by the World Health Organization (WHO) in 2021 [5], PEM is classified into three severity grades: mild (10–25%), moderate (25–39%), and severe (>40%). Common symptoms include immune dysfunction, deficiencies in vitamins and minerals, weight loss, growth retardation, and psychiatric disorders. Addressing and understanding malnutrition, particularly in children, is essential for improving public health outcomes [6]. Innovative and targeted interventions are required to overcome malnutrition [7].

Functional foods and dietary supplements, particularly probiotics, play a crucial role in addressing malnutrition. Several probiotic strains, particularly members of the genera *Lactobacillus* and *Bacillus*, have been reported to enhance digestive enzyme activity, including proteases, amylases, and lipases, thereby improving protein digestion and nutrient bioavailability (e.g., *Lactobacillus plantarum*, *L. rhamnosus*, *Bacillus subtilis*, *B. licheniformis*) [8–10]. These strains work synergistically to restore gut microbiota balance and promote better nutrient assimilation. Probiotic strains, including lactic acid bacteria (LAB), are commonly derived from sources such as the gastrointestinal tract (GIT), feces, and milk. These strains are widely recognized for their health-promoting properties and their role in maintaining gut microbiota balance [10, 11]. Effective strategies including the use of probiotic strains, can mitigate the severe impacts of malnutrition, fostering healthier generations [12].

Childhood undernutrition profoundly impacts the development of gut microbiota, leading to metabolic dysregulation. The gut microbiota, a complex community of microorganisms residing in the gut, significantly influences digestion, nutrient absorption, and immune function, impacting overall health. During early childhood, when the gut microbiota is still forming, undernutrition can alter its composition, reducing microbial diversity and beneficial species such as *Lactobacillus* and *Bifidobacterium*. Dysbiosis, an imbalance in the gut microbiota, can impair the gut's ability to efficiently break down and absorb nutrients. This disruption can lead to decreased production of essential microbial metabolites, such as short-chain fatty acids (SCFAs) [13, 14]. A healthy gut microbiota aids in synthesizing and absorbing essential nutrients, producing SCFAs that provide energy to colon cells, strengthen the intestinal mucosal barrier, protect against harmful pathogens, and support immune system maturation [15].

Over the past decade, probiotics have shown significant potential in addressing protein-energy malnutrition (PEM) by enhancing gastrointestinal health and aiding the recovery of immune function [16, 17]. Combining multiple strains to replicate the diverse composition of healthy gut microbiota may offer greater efficacy in addressing malnutrition and restoring gut health [18, 19]. Diverse bacterial strains in richer consortia are more likely to establish in new environments, enhancing nutrient uptake and resistance to pathogens [20, 21]. Numerous clinical trials have focused on evaluating single probiotic strains; however, research investigating MSPC and their impact on malnutrition remains limited. The key challenge is identifying synergistic probiotic strains for optimal benefits, as bacillus-based MSPCs offer enhanced pathogen

resistance and health outcomes. Targeted combinations of locally isolated strains, especially from dairy products, may outperform single strains [22–24].

*Bacillus* species were selected for this study due to their spore-forming nature, resistance to harsh gastrointestinal conditions, and capacity to produce extracellular enzymes involved in protein digestion, characteristics that make them suitable candidates for nutritional interventions in protein-energy malnutrition. Building on these attributes, the present work represents a systematic effort to formulate a Bacillus-based multi-strain probiotic consortium and to evaluate its *in-vitro* functional properties relevant to protein-energy malnutrition, including enzymatic activity, antioxidant potential, anti-inflammatory effects, and inter-strain compatibility.

## Materials and methods

### Materials

Media components and some other chemicals like De Man-Rogosa-Sharpe agar (MRS) agar, casein, bovine albumin fraction, and phosphate buffer were procured from HiMedia laboratories. Other chemicals like trichloroacetic acid ($CCl_3COOH$), 2,2'-azino-bis(3-ethyl benzo thiazoline-6-sulfonic acid (ABTS), potassium persulfate ($K_2S_2O_8$), 2,2-diphenyl-1-picrylhydrazyl (DPPH), and iron(III) chloride ($FeCl_3$) were acquired from SRL Pvt Ltd. 2,4,6-tri(2-pyridyl)-s-triazine (TPTZ) was purchased from Central Drug House Pvt Ltd and Iron(II) sulfate heptahydrate ($FeSO_4.7H_2O$) was from Oxford chemicals. Sodium carbonate ($Na_2CO_3$), Folin-Ciocalteu's phenol reagent, and ascorbic acid were supplied from MolyChem. Aspirin was obtained from Ecosprin and Methanol was from Renchem.

### Isolates

In the present study, bacterial isolates were obtained from Ms. Ishita Modasiya [25], who conducted the isolation, primary and secondary attribute evaluations, and probiotic attribute validations of these strains as part of her research [25]. The initial screening of these isolates included morphological examinations, Gram staining, negative staining, as well as oxidase and catalase tests, with the results summarized in S1 Table. Further evaluations were carried out to assess their acid and bile tolerance, cell auto-aggregation, and cell surface hydrophobicity, with the outcomes outlined in S2 Table and data of antioxidant activity shown in S3 Table.

### Antagonistic activity by spot method

The antagonistic activity of the selected 23 probiotic cultures (listed in Table 1) was assessed using the spot method on the MRS agar plate. Individual cultures were inoculated in MRS broth @1% (~$1x10^8$ CFU/mL) and kept at 37°C for 24 h in an anaerobic jar. Post-incubation, growth was compared with the 0.5 McFarland standard, which equates to $1.5 x 10^8$ CFU/mL. From the overnight grown cultures, 100 µL of each culture was individually spread on MRS agar plates, followed by placing a spot of 2 µL of the remaining 22 cultures. Plates were incubated for 24 h at 37°C under anaerobic conditions and zones of inhibition were observed to determine antagonistic interactions [26, 27].

### Proteolytic activity

The proteolytic activity of cell-free extract (CFE) obtained from individual isolates was evaluated using casein as a substrate. These isolates were later used for the multi-strain consortium formulation. A reaction mixture containing 500 µL of 1% (w/v) casein in 50 mM phosphate buffer (pH 7) and 200 µL of cell-free extract was incubated at 40 °C for 20 minutes. The reaction was terminated by adding 1 mL of 10% (w/v) trichloroacetic acid (TCA) and incubating the mixture at room temperature for 15 minutes. Unreacted casein was removed by centrifugation at 10,000 rpm for 5 minutes at room temperature. The supernatant (1 mL) was mixed with 2.5 mL of 0.44 M $Na_2CO_3$ and 1 mL of a three-fold diluted Folin-Ciocalteu's phenol reagent. The mixture was incubated in the dark at room temperature for 30 minutes, and the absorbance was measured at 660 nm against a reagent

**Table 1. Results of proteolytic activity and anti-inflammatory activity of selected 23 isolates.**

| Sr. no. | Isolates no. | Proteolytic activity (U/mL) | Anti-inflammatory activity (U/mL) |
|---|---|---|---|
| | *Lactiplantibacillus plantarum* NCDC 347 | 0.51 ± 0.01 | 3.11 ± 0.04 |
| | *Lacticaseibacillus rhamnosus* NDRI 184 | 0.35 ± 0.01 | 5.12 ± 0.01 |
| 1 | **PIG5 CI** | **0.61 ± 0.01** | **0.11 ± 0.02** |
| 2 | **PIG3IR** | **0.36 ± 0.01** | **0.59 ± 0.06** |
| 3 | PIG6IR | 0.36 ± 0.01 | 0.19 ± 0.02 |
| 4 | PIB13MR | 0.38 ± 0.02 | 0.32 ± 0.03 |
| 5 | PIB14TR | 0.39 ± 0.02 | 3.08 ± 0.04 |
| 6 | PIB12FI | 0.37 ± 0.02 | 0.12 ± 0.01 |
| 7 | PIB12RB | 0.40 ± 0.02 | 0.16 ± 0.01 |
| 8 | PIM10FI | 0.47 ± 0.01 | 0.18 ± 0.04 |
| 9 | PIY1RC | 0.44 ± 0.05 | 0.12 ± 0.02 |
| 10 | PIC20SC | 0.42 ± 0.02 | 0.08 ± 0.01 |
| 11 | PIC20SY | 0.36 ± 0.01 | 0.24 ± 0.01 |
| 12 | PIC23R | 0.36 ± 0.04 | 0.06 ± 0.01 |
| 13 | PIC22IF | 0.37 ± 0.02 | 0.12 ± 0.02 |
| 14 | PIC22RI | 0.36 ± 0.01 | 0.09 ± 0.02 |
| 15 | PIM8CR | 0.35 ± 0.03 | 0.05 ± 0.01 |
| 16 | PIC5CR | 0.35 ± 0.04 | 0.08 ± 0.01 |
| 17 | PIM9FI | 0.41 ± 0.02 | 0.27 ± 0.01 |
| 18 | **PIM9CR** | **0.37 ± 0.03** | **0.45 ± 0.08** |
| 19 | PIB9SR | 0.33 ± 0.01 | 0.08 ± 0.01 |
| 20 | PIB10CR | 0.35 ± 0.01 | 0.16 ± 0.04 |
| 21 | PIB10MR | 0.33 ± 0.02 | 0.20 ± 0.05 |
| 22 | PIB9MR | 0.35 ± 0.01 | 0.05 ± 0.01 |
| 23 | PIB9CR | 0.36 ± 0.01 | 0.69 ± 0.02 |
| | **MSPC** | **0.52 ± 0.03** | **5.33 ± 0.02** |

All experiments were performed in technical triplicates, and results are expressed as mean ± SD.

blank using a tyrosine standard curve. The concentration of reaction products was calculated using the linear regression equation obtained from the standard curve (y = mx + c), where absorbance (y) was proportional to concentration (x) [28, 29].

## Anti-inflammatory activity by albumin denaturation assay

The anti-inflammatory activity of the isolates was assessed using the albumin denaturation inhibition assay. A reaction mixture containing 500 µL/mL of culture supernatant and 1% bovine serum albumin (BSA) solution was prepared. Similarly, a standard mixture containing 1 mg/mL aspirin was prepared. Both mixtures were incubated at room temperature for 20 minutes, followed by heating at 51°C for 20 minutes to induce protein denaturation. The mixtures were then cooled, and the turbidity of the albumin protein was measured at 660 nm using a Shimadzu-1900 UV-visible spectrophotometer. The concentration of reaction products was calculated using the linear regression equation obtained from the standard curve (y = mx + c), where absorbance (y) was proportional to concentration (x) [30].

## Antioxidant activities

**ABTS scavenging activity.** The antioxidant activity of the isolates was determined using the ABTS radical scavenging assay, with modifications based on the method described by Pieniz (2014). The ABTS radical cation was generated by

reacting a stock solution of ABTS with 2.45 mM $K_2S_2O_8$ (final concentration) and allowing the mixture to stand in the dark at room temperature for 16 hours. The ABTS solution was then diluted with 1X PBS to obtain an absorbance of $0.700 \pm 0.020$ at 734 nm. Ascorbic acid (100–1000 µM) was used as a standard. To assess the antioxidant activity, 10 µL of cell-free supernatant (or standard) was added to 990 µL of the ABTS solution. The absorbance was measured at 734 nm at 30-second intervals for 6 minutes. The culture medium was used as a control. The percentage inhibition of free radicals was calculated using an ascorbic acid standard curve. The initial and final optical density (OD) values were used to calculate the percentage of radical scavenging activity using the following formula: % scavenging activity = $(A_{734}\text{Initial}-A_{734}\text{Final})/A_{734}\text{Initial} \times 100$ [31].

**DPPH scavenging activity.** The DPPH method relies on the reduction of the stable DPPH radical by antioxidants, resulting in a decrease in absorbance at 515 nm. A 5 mM DPPH solution was prepared in 70% methanol, stored in an amber bottle, and kept in the dark. Before use, the absorbance of the DPPH solution was adjusted to $0.70 \pm 0.02$ at 515 nm. Twenty microliters of the test samples (culture-free supernatant) or standard (ascorbic acid, 100–1000 µM) were added to separate tubes containing 980 µL of DPPH solution. The mixtures were homogenized by shaking, and the absorbance was measured at 30-second intervals for 6 minutes at 515 nm. A negative control containing 2 µL of 70% methanol and 980 µL of DPPH solution was included. All experiments were performed in triplicate. The percentage inhibition of free radicals was calculated using the following equation: % scavenging activity = $(A_{515} \text{Initial}- A_{515} \text{Final}/A_{515} \text{Initial}) \times 100$ [31].

**FRAP (ferric reducing antioxidant potential) activity.** The reducing power of the cell-free supernatant (CFS) was measured by assessing its ability to reduce $Fe^{3+}(CN)_6$ to $Fe^{2+}(CN)_6$, as described by Shori et al. (2022). A FRAP reagent containing 300 mM acetate buffer, 8 mM 2,4,6-tri(2-pyridyl)-s-triazine (TPTZ), and 20 mM $FeCl_3$ was mixed with 400 µL of CFS or $FeSO_4 \cdot 7H_2O$ standard solutions (0.3–1.0 µg/mL) in a 10:1:1 ratio. The mixture was incubated in a water bath at 37°C for 10 minutes, and the absorbance was measured at 593 nm using a Shimadzu-1900 UV-visible spectrophotometer. The reducing power was calculated by comparing the absorbance of the sample to the $FeSO_4$ standard curve (Al Zahrani and Shori 2023), using the following formula: Reducing Power = $(A_{593}/$ Slope of the $FeSO_4$ standard curve $\times$ Dilution factor) [32, 33].

## Formulation of MSPC

The formulation of the multi-strain probiotic consortium was based on the results obtained from assessing the antagonistic, proteolytic, anti-inflammatory, and antioxidant activities of 23 different isolates. After initial screening, three isolates were selected based on their superior performance in these activities. The growth analysis of these isolates was carried out to evaluate their compatibility and potential synergistic effects when combined. Additionally, the MSPC was subjected to further analysis to confirm its proteolytic, anti-inflammatory, and antioxidant properties, following the same protocols outlined for individual isolates. Equal volumes of overnight cultures of each selected strain (PIG5 CI, PIG3IR, PIM9CR), each adjusted to $\sim 1 \times 10^8$ CFU/mL, were combined aseptically to prepare the MSPC. The consortium was then cultured in MRS broth at 37°C under anaerobic conditions for 24 h before use in assays.

## Microbial growth analysis

Selected isolates namely *Bacillus spizizenii* (PIG5 CI; Accession no. OP846620), *Bacillus tequilensis* (PIG3IR; Accession no. PP237761), and *Bacillus rugosus* (PIM9CR; Accession no. PP237763). Taxonomic identification of the selected strains (PIG5 CI, PIG3IR, PIM9CR) was performed by 16S rRNA gene sequencing using universal primers 27F and 1492R. The sequences were compared using NCBI BLAST, and identities were confirmed with ≥98% similarity. Sequences were deposited in GenBank under accession numbers OP846620, PP237761, and PP237763 respectively [25]. Also, 2 standard cultures *Lactiplantibacillus plantarum* NCDC 347 (*L. plantarum* NCDC 347) and *Lacticaseibacillus rhamnosus* NDRI 184 (*L. rhamnosus* NDRI 184) were individually inoculated in MRS broth at 1% ($\sim 1 \times 10^8$ CFU/mL) and kept at 37 ˚C for 24

h. Post incubation after reaching an optical density (OD) of 0.1 at 600 nm, 300 μL of each culture was inoculated into 30 mL of sterile MRS broth and incubated at 37 ˚C for up to 48 hours. The OD at 600 nm was measured at predefined intervals (0, 1, 3, 5, 7, 24, and 48 hours) to monitor the growth kinetics of each isolate.

### Synergistic activity: Cell-cell free extract interaction analysis

Based on the findings of antagonistic activity, proteolytic activity, and anti-inflammatory attributes, selected cultures were subjected to synergistic activity to evaluate the growth-supportive potential of CFE of one isolate on another cell. In this method, MRS medium pre-inoculated with viable cells of one isolate (suspension OD equivalent to 0.5 McFarland standard that gives ~ 1x10^8 CFU/mL) was added with the CFE of another isolate and incubated together to check the synergistic activity. The CFE preparation and absorbance measurement was performed as mentioned by Maniya et al. (2024). A comparison of the individual isolate's growth curve and supernatant of selected isolates for the development of MSPC was plotted [27].

### Ethics statement

This study involved in-vitro experiments only and did not include human participants or animals. Therefore, ethical approval was not required.

### Statistical analysis

Quantitative data obtained for each experiment, that were performed in triplicates, was added into the Microsoft Excel 2019. The raw data was subjected to the calculation of mean and standard deviation. Microsoft Excel was also used to generate graphical representations of suitable data sets.

## Results

**MSPC:** *Bacillus spizizenii* BAB 7915 (PIG5 CI), *Bacillus tequilensis* (PIG3IR), and *Bacillus rugosus* (PIM9CR).

### Antagonistic activity

Antagonistic activity refers to the ability of one isolate to strongly inhibit the growth of another, observed as a clear zone of inhibition around a microbial culture spot on MRS agar plates. Our study identified seven isolates (PIB14TR, PIB12FI, PIM10FI, PIY1RC, PIG5 CI, PIB13MR, and PIB12RB) that demonstrated strong inhibitory effects on the growth of others, as illustrated in the heatmap in Fig 1. Conversely, some isolates, such as PIC20SC, PIC20SY, PIC22RI, and PIM9FI, exhibited overgrowth when co-cultivated, indicating their competitive advantage in growth. Fig 1 visually highlights these specific interactions among the 23 isolates tested. The rest of the cultures, which did not show significant antagonistic activity, were considered suitable for further analysis and potential applications.

Having established that these three Bacillus strains can coexist without mutual antagonism, we next sought to evaluate their collective metabolic efficiency, specifically focusing on their ability to degrade complex proteins, a critical requirement for PEM intervention.

### Proteolytic activity

Out of 23 isolates tested for proteolytic attribute, 12 cultures demonstrated the potential to degrade casein. They showed proteolytic activity greater than 0.36 U/mL, (data presented in Table 1). Among the selected cultures for the consortium, PIG5 CI demonstrated the highest proteolytic activity with values of 0.61 ± 0.01 U/mL. While, PIM9CR and PIG3IR showed proteolytic activity of 0.37 ± 0.03 U/mL and 0.36 ± 0.01 U/mL, respectively. The standard cultures, *L. rhamnosus* NDRI 184 and *L. plantarum* NCDC 347 exhibited 0.35 ± 0.01 U/mL and 0.51 ± 0.01 U/mL, respectively. The MSPC has shown

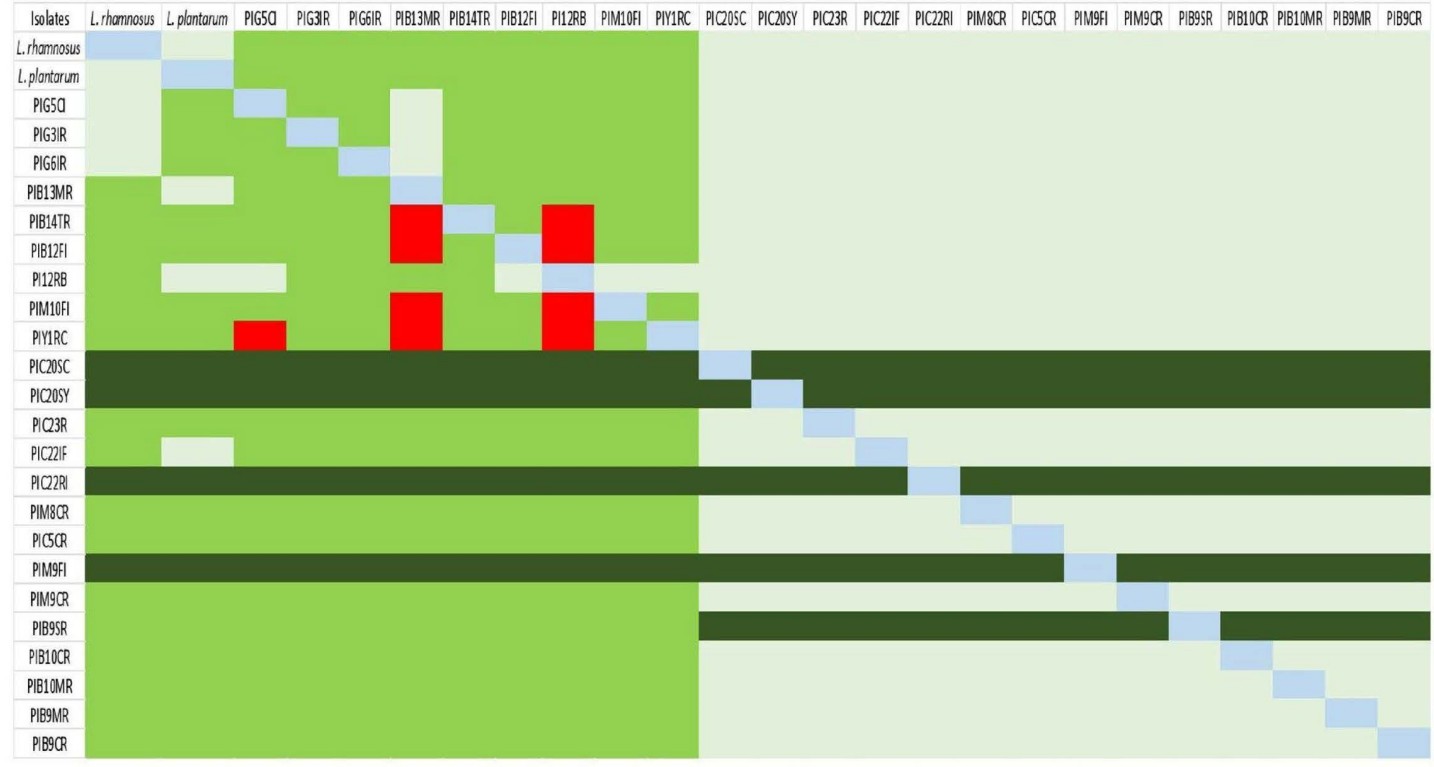

**Fig 1. Heat map representing the antagonistic activity of selected 23 isolates against each other.** Footnote: Red (Zone of inhibition of growth), Green (test organism showing visible growth), Dark green (Overgrowth), of test organism), Off-white (No growth), Light blue (N/A).

proteolytic activity of 0.52±0.03 U/mL. Having proteolytic potential makes the probiotic isolates a potential candidate for inclusion in the MSPC as the bioactive peptides produced by the isolates may contribute to immune modulation and improved gut health, aiding to recovery from PEM. Incorporating such strains into a MSPC could provide a synergistic effect, targeting protein digestion and gut microbiota balance, intestinal health, and systemic inflammation, which are often disrupted in PEM.

While efficient nutrient breakdown is essential, the therapeutic value of a probiotic in PEM also depends on its ability to survive and protect the host environment; therefore, the consortium was further evaluated for its capacity to neutralize oxidative stress and suppress inflammatory markers.

### Anti-inflammatory assay

In the anti-inflammatory assay, 15 out of the 23 tested cultures demonstrated activity at concentrations above 0.1 mg/mL, as shown in Table 1. Among the selected strains, PIG5 CI, PIG3IR, and PIM9CR displayed moderate anti-inflammatory potential, with activity levels of 0.11±0.02 U/mL, 0.59±0.06 U/mL, and 0.45±0.08 U/mL, respectively. These results suggest their suitability for further applications. In comparison, the standard strains *L. rhamnosus* NDRI 184 and *L. plantarum* NCDC 347 exhibited higher anti-inflammatory activity, with concentrations of 5.12±0.01 U/mL and 3.11±0.04 U/mL, respectively. The anti-inflammatory activity of MSPC was found to be 5.33±0.02 U/mL. These results confirm the superior anti-inflammatory potential of these standard strains, highlighting their role as benchmarks for evaluating the efficacy of other cultures.

## Antioxidant Activity

The antioxidant properties of all 23 isolates were assessed using ABTS, DPPH, and FRAP assays, with results detailed in Table S3. In the ABTS radical scavenging assay, isolates PIM9CR, PIG3IR, and PIG5 CI exhibited notable antioxidant activity at 84.39±0.59%, 82.51±0.22%, 82.09±0.52%, respectively, as shown in Fig 2. Among individual isolates, MSPC showed relatively higher activity with a value of 86.25±0.37% reduction. Standard strains *L. rhamnosus* NDRI 184 and *L. plantarum* NCDC 347 demonstrated even higher ABTS scavenging activities, with reductions of 88.92±0.92% and 89.37±0.45%, respectively. In the DPPH assay, PIG5 CI, PIG3IR, PIM9CR, and MSPC achieved reductions of 20.16±1.34%, 32.95±1.7%, 27.18±0.87%, and 47.11±2.0%, respectively (Fig 2). Notably, *L. rhamnosus* NDRI 184 and *L. plantarum* NCDC 347 recorded the highest activity, with reductions of 43.45±4.63% and 43.32±0.23%, respectively. The FRAP assay, which evaluates ferric-reducing antioxidant power, revealed reductions of 80.93±0.12%, 83.66±0.05%, 86.14±0.03%, and 82.95±0.61% for PIG5 CI, PIG3IR, PIM9CR, and MSPC, respectively (Fig 2). In contrast, the standard strains showed varying activity, with *L. rhamnosus* NDRI 184 achieving 78.18±0.08% and *L. plantarum* NCDC 347 showing 65.40±0.41%.

## Microbial growth analysis

The growth dynamics of three selected isolates (PIG5 CI, PIG3IR, and PIM9CR) and two standard probiotic strains, *L. rhamnosus* NDRI 184 and *L. plantarum* NCDC 347, were evaluated and compared, with the results shown in Fig 3. Initially, PIG5 CI, PIG3IR, and PIM9CR were in the lag phase up to 7 h and all selected isolates entered the log phase after 7 h. It was observed that standard cultures *L. rhamnosus* NDRI 184 and *L. plantarum* NCDC 347 entered the exponential phase after 5 h. The growth measurements obtained after 24 and 48 h indicated that the isolates achieved the stationary phase. This growth curve data highlights differences in the growth kinetics between standard and selected isolates and will be instrumental in analyzing synergistic activity.

## Cell-cell free extract interaction analysis

To evaluate synergistic activity, the interaction of PIG5 CI cells with the cell-free supernatants of other isolates was assessed, as shown in Fig 4A. The cells of PIG5 CI entered the log phase after 7 h when grown with the supernatants of

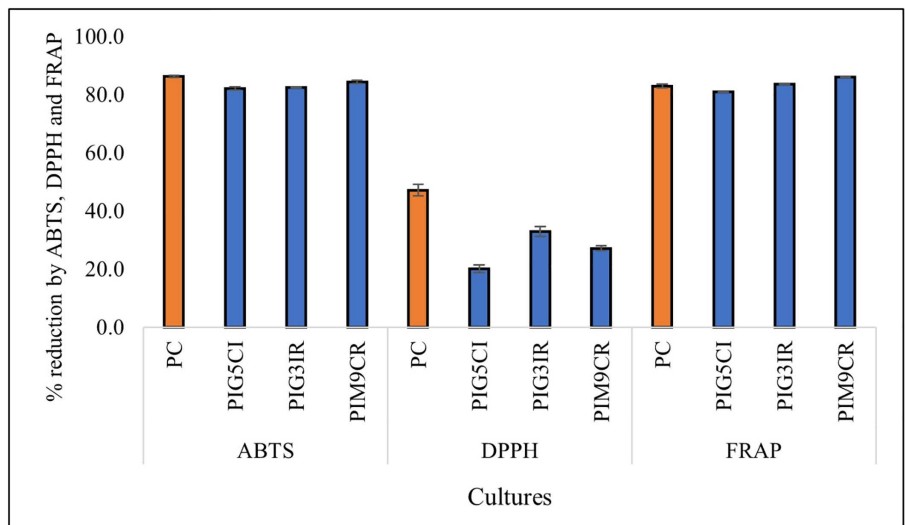

**Fig 2. Percent Inhibition of ABTS, DPPH, and FRAP free radical by antioxidant method of MSPC (PIG5 CI, PIG3IR, PIM9CR).** Note: Error bars represent mean±standard deviation (SD) of triplicate experiments.

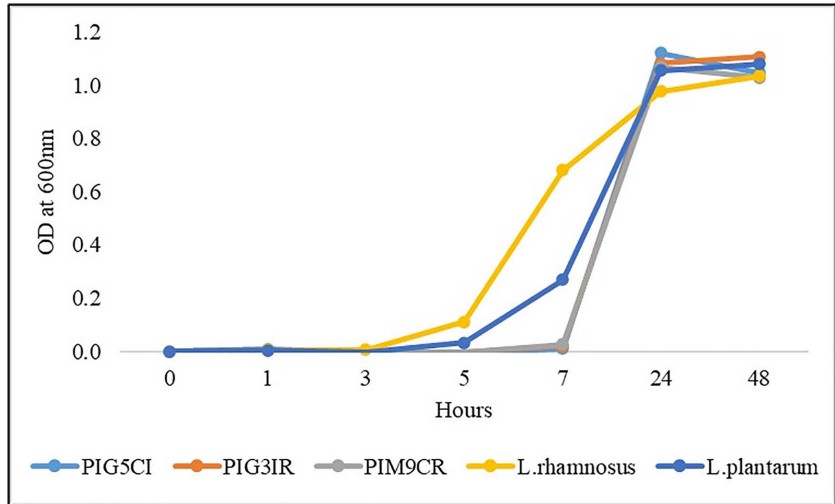

**Fig 3. Growth curve of selected bacterial and standard cultures.**

PIG3IR, PIM9CR, and *L. plantarum* NCDC 347. However, in the presence of the supernatant from *L. rhamnosus* NDRI 184, the log phase was reached within 5 h, indicating enhanced growth. PIG3IR entered the log phase at 7 h when exposed to the supernatants of PIG5 CI, PIM9CR, and *L. plantarum* NCDC 347, while it reached the log phase faster, within 5 h, with the supernatant of *L. rhamnosus* NDRI 184, indicating enhanced growth, as shown in Fig 4B. Fig 4C shows that PIM9CR, PIG5 CI, and PIG3IR entered the log phase after 7 h, while *L. rhamnosus* NDRI 184 and *L. plantarum* NCDC 347 entered earlier, at 5 h, suggesting faster growth.

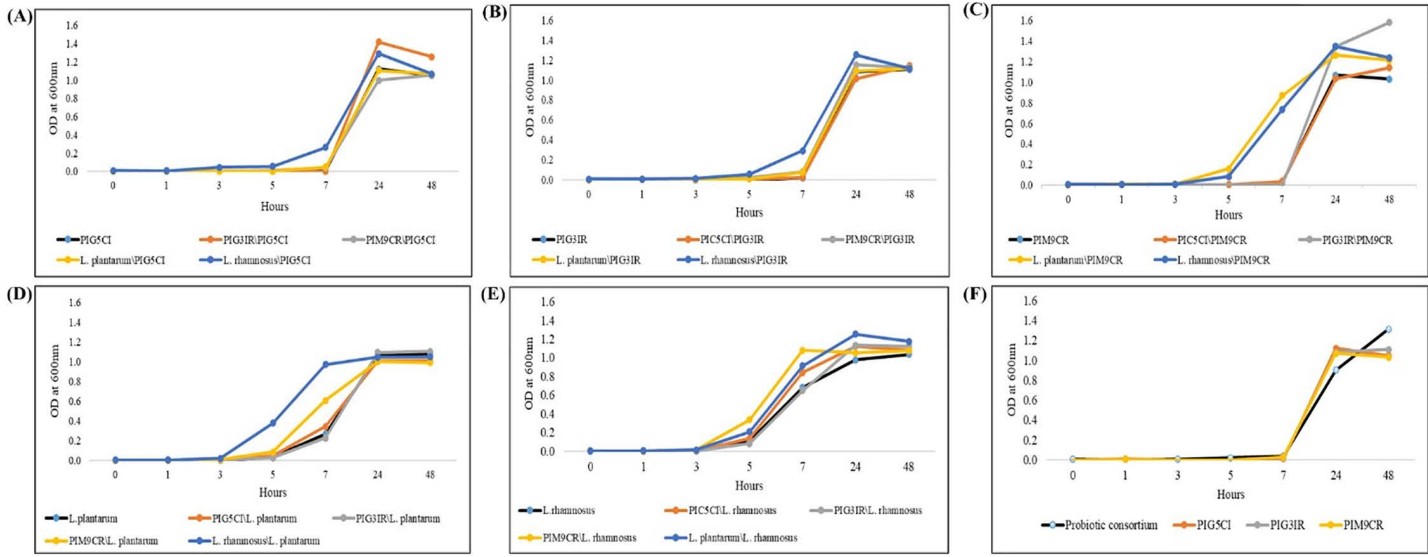

**Fig 4. Synergistic activity of PIG5 CI, PIG3IR, PIM9CR, *L. plantarum, L. rhamnosus,* and probiotic consortium.** Footnote: **(A)** Effect of CFEs on cells of PIG5 CI, **(B)** Effect of CFEs on cells of PIG3IR, **(C)** Effect of CFEs on cells of PIM9CR, **(D)** Effect of CFEs on cells of *L. plantarum,* **(E)** Effect of CFEs on cells of *L. rhamnosus,* **(F)** Synergistic activity of probiotic consortium with individual cultures.

In Fig 4D, *L. plantarum* NCDC 347 entered the log phase at 5 hours along with PIG5 CI, PIG3IR, and PIM9CR, but at 3 h when paired with *L. rhamnosus* NDRI 184, indicating accelerated growth. Similarly, in Fig 4E, *L. rhamnosus* NDRI 184 entered the log phase at 5 h with the selected isolates, but at 3 h with *L. plantarum* NCDC 347, showing enhanced growth. All selected cultures for the MSPC, including PIG5 CI, PIG3IR, PIM9CR, and the probiotic consortium (PC), entered the log phase at 7 h, as shown in Fig 4F. After 24 h, PIG5 CI showed maximum growth with the PIG3IR supernatant, while PIG3IR exhibited the highest growth with *L. rhamnosus* NDRI 184. Growth patterns varied, with *L. plantarum* NCDC 347 supernatant promoting substantial growth across isolates. By 48 h, all selected strains reached the stationary phase, while the MSPC remained in the log phase.

Collectively, these in-vitro data provide a comprehensive profile of the MSPC's functional capabilities, setting the stage for a detailed interpretation of how these synergistic properties might translate into clinical benefits for malnourished populations.

## Discussion

The development of a robust Bacillus-based multi-strain probiotic consortium (MSPC) represents a strategic shift from single-strain interventions toward replicating the functional diversity of a healthy gut microbiota. In this study, inter-strain compatibility served as the foundational pillar; by excluding antagonistic isolates and prioritizing those with synergistic growth patterns, we ensured the structural stability and functional consistency required for a viable formulation. This rational selection process directly informed the consortium's efficacy, most notably its significant proteolytic activity ($0.52 \pm 0.03$ U/mL). Given that impaired protein digestion is a hallmark of protein-energy malnutrition, the ability of these Bacillus-derived proteases to hydrolyze dietary proteins into absorbable peptides suggests that this MSPC is not merely a probiotic supplement but a targeted metabolic intervention for nutritional rehabilitation.

Antagonistic activity was initially assessed as a screening criterion to evaluate inter-strain compatibility rather than to infer pathogen inhibition *in-vivo*. Isolates exhibiting inhibitory interactions were excluded from consortium development to avoid mutual growth suppression that could compromise formulation stability. Only non-antagonistic strains were subsequently evaluated for synergistic growth behavior and functional attributes. This stepwise selection strategy ensured that the final consortium comprised compatible strains capable of coexisting without inhibitory effects, a critical requirement for the development of a stable and effective multi-strain probiotic formulation. To ensure optimal consortium performance, antagonistic interactions among the selected strains were assessed. Strains exhibiting inhibitory effects, such as PIB14TR, PIB12RB, PIM10FI, and PIY1RC, were excluded from the consortium. The remaining three strains demonstrated compatibility and even exhibited synergistic growth, reinforcing their suitability for co-administration. The synergistic potential of PIG5 CI has been previously reported by [27], who also utilized this strain in the development of probiotic consortia. Our findings further corroborate the beneficial role of PIG5 CI in promoting gut health and overall well-being, particularly in the context of PEM [27].

Proteolytic activity is a particularly relevant functional trait for addressing protein-energy malnutrition, as inadequate protein digestion and utilization are central features of the condition. The strong proteolytic potential observed in the consortium suggests a capacity to enhance the breakdown of dietary proteins into absorbable peptides and amino acids. Previous studies have highlighted the role of Bacillus-derived proteases in improving protein digestibility, supporting the functional relevance of the consortium in a nutritional context. Although the proteolytic activity of the MSPC was lower than that of the single strain PIG5 CI, the consortium was designed to provide functional synergy through complementary enzymatic activities, antioxidant capacity, anti-inflammatory effects, and improved strain compatibility, rather than maximizing a single functional parameter. Previous studies have demonstrated the proteolytic potential of various bacterial strains. Lactic acid bacteria (LAB) possess a highly complex proteolytic system comprising three major components: cell wall-associated proteinases that initiate casein hydrolysis into oligopeptides, specialized transport systems that import these peptides into the cell, and a diverse set of intracellular peptidases that further break them down into smaller

peptides and free amino acids. Although genome-based analyses have provided insights into these proteolytic elements, the system's intricacy continues to pose challenges for the isolation and characterization of many of its functional proteins, which remain only partially understood [29]. Sony and Potty (2016) reported that *Proteus mirabilis* TKMFT19 exhibited a high proteolytic activity of 126.6 ± 1.98 mg/mL [28]. Similarly, Gracia-Cano et al. (2019) identified *Lactobacillus casei* (*L. casei* OSU-PECh-C), *Lactobacillus paracasei* (*L. paracasei* OSU-PECh-BA), *Lactobacillus plantarum* (*L. plantarum* OSU PECh-A and *L. plantarum* OSU-PECh-BB) as strong proteolytic strains from dairy products [34]. Authors further highlighted the proteolytic activity of *Lactobacillus fermentum* NMCC-14, which displayed a high level of 28.5 ± 0.1 U/mg [35].

Proteolytic activity has been widely reported among probiotic microorganisms. For instance, strains such as Proteus mirabilis, Lactobacillus casei, and Lactobacillus fermentum have demonstrated the ability to secrete extracellular proteases that facilitate protein degradation. In the present study, the proteolytic activity observed in the selected Bacillus strains supports their potential role in improving protein hydrolysis and nutrient availability. Such functionality is particularly relevant in the context of protein-energy malnutrition, where impaired protein digestion and absorption contribute to nutritional deficiencies. Therefore, the incorporation of proteolytically active probiotic strains within a consortium may enhance dietary protein utilization and support host nutritional recovery.

Chronic low-grade inflammation is increasingly recognized as a contributing factor in the pathophysiology of protein-energy malnutrition, impairing nutrient absorption and metabolic efficiency. The observed anti-inflammatory activity of the consortium suggests a potential role in modulating inflammatory responses that may otherwise exacerbate malnutrition. The anti-inflammatory potential of our MSPC is a critical factor in its ability to ameliorate PEM. The proteolytic activity threshold of PIG3IR (0.36 Units/mL) was selected because the corresponding strain also exhibited the highest anti-inflammatory activity among the tested isolates. Strain selection for consortium development was therefore based on combined functional performance rather than maximal proteolytic activity alone. Probiotic strains within the consortium can modulate the inflammatory response, promoting gut health and overall well-being. The observed inhibition of albumin denaturation by the multi-strain probiotic consortium and individual isolates suggests their potential to prevent protein destabilization during inflammatory conditions. This protective effect likely stems from bioactive compounds in the bacterial supernatants that interact with albumin, thereby reducing the availability of denatured, immunogenic proteins that trigger inflammatory cascades. This mechanism is closely linked to the prevention of albumin denaturation, a well-established biomarker of inflammation and cellular damage. A stable albumin profile reflects reduced inflammatory activity, which is pivotal in mitigating PEM. The authors reported a 61.6% anti-inflammatory activity in *Lactobacillus agilis* compared to 69.0% for aspirin in an albumin denaturation assay [30]. While their study provides valuable insights, our approach of quantifying anti-inflammatory activity in terms of concentration offers greater specificity and allows for more precise comparisons across different strains and experimental conditions. Aguilar-Toalá et al. (2017) demonstrated anti-inflammatory activity in *Lactobacillus plantarum* within a range of 55.61 to 17.43 µg/mL [36]. This variability underscores the importance of strain-specific effects and the need for rigorous screening to identify the most potent probiotics for PEM intervention.

Oxidative stress is frequently elevated in individuals with protein-energy malnutrition and contributes to tissue damage and metabolic dysfunction. The antioxidant activity demonstrated by the consortium across multiple assays indicates its potential to counteract oxidative stress through different mechanisms, including radical scavenging and reducing power. These findings are consistent with earlier reports describing antioxidant metabolite production by Bacillus species, further supporting the functional robustness of the consortium. The antioxidant properties exhibited by our probiotic isolates and the MSPC are crucial for mitigating oxidative stress, a major contributor to various diseases, including malnutrition. Our findings demonstrate superior antioxidant activity compared to several well-established probiotic strains. The ABTS radical scavenging activity of our isolates and MSPC, surpassing the results reported by [37], underscores their potent antioxidant potential [37]. Similarly, the DPPH radical scavenging activity of our consortium outperforms the values reported by [37, 38]. These results highlight the potential of our MSPC to neutralize harmful free radicals and protect cells from oxidative damage. Furthermore, the elevated FRAP activity of our isolates and MSPC, compared to *Lactococcus lactis* subsp.

*cremoris* LL95 [39], indicates their strong reducing power. This enhanced ability to donate electrons to reduce oxidized compounds suggests a robust defense mechanism against oxidative stress. The production of bioactive compounds, such as peptides, exopolysaccharides, and antioxidant enzymes, by our probiotic isolates and MSPC further contributes to their antioxidant properties. These compounds can effectively scavenge ROS, reduce ascorbate autoxidation, and chelate metal ions, as evidenced by studies like [40] and [41]. By mitigating oxidative stress, our probiotic consortium can potentially alleviate the negative impacts of PEM, which is often associated with increased oxidative damage.

The absence of antagonism and the observed synergistic growth patterns among the constituent strains are critical for the stability and functional consistency of a multi-strain probiotic formulation. Incompatible strain interactions can compromise efficacy and shelf-life; therefore, the demonstrated compatibility supports the feasibility of developing the consortium as a unified formulation. Similar compatibility-driven approaches have been emphasized in successful multi-strain probiotic designs. The synergistic activity observed among the probiotic isolates in the consortium is a key factor contributing to its enhanced efficacy. This phenomenon, whereby multiple strains work together to achieve greater benefits than when used individually, is well-documented in probiotic research. The authors have previously demonstrated the synergistic growth patterns of various bacterial combinations, including *L. paracasei* and *P. prevotii*. Our findings align with these observations, as the MSPC exhibited superior growth characteristics compared to single-strain cultures [42]. Also, the growth curve analysis was primarily designed to assess early growth dynamics up to 7 h. Additional measurements at 24 h and 48 h were included to confirm the onset and persistence of the stationary phase. However, the absence of intermediate sampling points between 7 h and 24 h limited detailed kinetic analysis of the exponential growth phase. Future studies will include more frequent time-point sampling to enable comprehensive growth kinetics evaluation. The prolonged stationary phase of the consortium suggests a more robust and stable microbial community, which may be advantageous for gut colonization and persistence. This increased microbial stability could potentially enhance the consortium's ability to modulate gut microbiota composition and function, thereby mitigating the negative effects of PEM. The specific mechanisms underlying the synergistic effects of the MSPC remain to be fully elucidated. However, it is likely that interspecies interactions, such as nutrient exchange and metabolite production, contribute to the observed benefits.

The use of a multi-strain probiotic consortium may offer advantages over individual strains due to potential complementary metabolic activities among the constituent microorganisms. Different Bacillus strains may contribute distinct enzymatic or metabolic functions, which together could enhance overall functional performance. Although the consortium did not consistently exceed the activity of the best individual strain in every assay, the combined presence of multiple functional traits may provide broader physiological benefits. Such complementary interactions are commonly reported in multi-strain probiotic formulations and may contribute to improved resilience and functional stability.

Further research is needed to explore these mechanisms in detail. Given the promising results of our *in-vitro* studies, the MSPC represents a potential therapeutic strategy for the management of PEM. Future clinical trials are warranted to evaluate the efficacy and safety of this consortium in human subjects. By targeting the gut microbiota, the MSPC may offer a novel approach to improving nutritional status and overall health in individuals suffering from PEM. While the present work represents the first phase of a multi-stage investigation aimed at developing a probiotic-based intervention for protein-energy malnutrition. This phase was designed to perform *in-vitro* functional screening and compatibility assessment of a Bacillus-based multi-strain probiotic consortium. As such, the study does not capture host-microbiome interactions, intestinal colonization dynamics, or systemic immune modulation. Subsequent phases of this research are planned to include *in-vivo* validation using appropriate animal models, followed by metagenomic analysis to elucidate microbiome-level alterations and functional pathways associated with consortium administration.

Overall, the findings of this study demonstrate that the selected Bacillus isolates and the formulated probiotic consortium possess several functional characteristics relevant to nutritional health, including proteolytic, antioxidant, and anti-inflammatory activities. These properties suggest that such probiotic formulations may have potential applications as functional dietary supplements aimed at supporting nutritional recovery in malnourished populations. However, additional

studies are required to confirm these effects *in-vivo* and to elucidate the mechanisms underlying the observed functional activities.

## Conclusion

This study concludes that the rational assembly of a *Bacillus*-based multi-strain consortium (MSPC) provides a synergistic functional advantage over single-strain probiotics in addressing the metabolic gaps of PEM. By prioritizing inter-strain compatibility, we developed a stable formulation capable of high-level protein degradation (0.52 U/mL) and significant inflammatory modulation (5.33 U/mL). These quantitative milestones are critical, as they directly address the proteolytic deficiency and mucosal inflammation associated with malnourished states. Our findings move beyond basic probiotic characterization to provide a targeted functional intervention. Consequently, this MSPC serves as a promising, shelf-stable candidate for further *in-vivo* trials, offering a scalable biotechnological solution to improve nutritional outcomes in PEM-affected populations.

## Supporting information

**S1 Table. Gram staining, negative staining, catalase and oxidase test results of selected 23 isolates with standard strains *Lactiplantibacillus plantarum* NCDC 347, *Lacticaseibacillus rhamnosus* NDRI 184.**
(DOCX)

**S2 Table. Secondary screening results of selected 23 isolates with standard strains *Lactiplantibacillus plantarum* NCDC 347, *Lacticaseibacillus rhamnosus* NDRI 184.**
(DOCX)

**S3 Table. Antioxidant activity of selected 23 isolates.**
(DOCX)

## Acknowledgments

We are thankful to the concerned authorities of RK University, School of Science, and School of Pharmacy for providing the required resources and support for the smooth conduct of the research work. We also thank the SHODH, Government of Gujarat, for providing fellowships to the scholars.

## Author contributions

**Conceptualization:** Priya Mori, Vijay Kumar.

**Data curation:** Hina Maniya.

**Formal analysis:** Priya Mori, Ishita Modasiya, Mehul Chauhan, Hina Maniya, Vijay Kumar, Apurba Kumar Sarkar.

**Funding acquisition:** Apurba Kumar Sarkar.

**Investigation:** Priya Mori.

**Methodology:** Priya Mori, Ishita Modasiya.

**Project administration:** Priya Mori.

**Resources:** Ishita Modasiya.

**Supervision:** Vijay Kumar.

**Validation:** Vijay Kumar, Apurba Kumar Sarkar.

**Visualization:** Priya Mori.

**Writing – original draft:** Priya Mori.

**Writing – review & editing:** Mehul Chauhan, Vijay Kumar, Apurba Kumar Sarkar.

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
