## [Decision Letter · Decision Letter 0]

8 Jan 2026

Dear Dr. Sarkar,

Thank you for submitting your manuscript to PLOS ONE. After careful consideration, we feel that it has merit but does not fully meet PLOS ONE’s publication criteria as it currently stands. Therefore, we invite you to submit a revised version of the manuscript that addresses the points raised during the review process.

We look forward to receiving your revised manuscript.

Kind regards,

Babak Pakbin

Academic Editor

PLOS One

Journal Requirements:

Reviewers' comments:

Reviewer's Responses to Questions

**Comments to the Author**

1. Is the manuscript technically sound, and do the data support the conclusions?

Reviewer #1: No

Reviewer #2: Partly

2. Has the statistical analysis been performed appropriately and rigorously?

Reviewer #1: I Don't Know

Reviewer #2: No

3. Have the authors made all data underlying the findings in their manuscript fully available?

Reviewer #1: Yes

Reviewer #2: Yes

4. Is the manuscript presented in an intelligible fashion and written in standard English?

Reviewer #1: Yes

Reviewer #2: No

Reviewer #1: Please see the attached review document for detailed comments. A summary of the main points is provided below.

This manuscript presents an in vitro evaluation of a multi-strain probiotic consortium composed of Bacillus spizizenii, Bacillus tequilensis, and Bacillus rugosus as a potential intervention for protein-energy malnutrition. The topic is timely and relevant, and the experimental workflow is logically structured from strain selection to functional evaluation. Key concerns include unclear statistical reporting, figure presentation issues, inconsistencies in data visualization, and limited discussion of methodological constraints. Substantial revision is required to improve scientific clarity, data interpretation, and alignment between the results and conclusions.

Reviewer #2: This study addresses a global health challenge, protein-energy malnutrition, by proposing a novel multi-strain probiotic consortium comprising Bacillus spizizenii, Bacillus tequilensis, and Bacillus rugosus. The study also reports screening outputs, including antagonism and compatibility, proteolysis, anti-inflammatory, and antioxidant assays. While the scientific concept is promising, the manuscript must be enhanced substantially in the scientific and mathematical reporting. Additionally, language quality, grammar, formatting, and figure captions require thorough revision. The principal remarks are listed below to help the authors revisit and improve the manuscript.

1. In the introduction section, when talking about malnutrition in India, supporting statistics needs to be added. Without any data, the claim in the study feels vague.

2. In the introduction section, in the 2nd paragraph, it has been reported that several probiotic strains have been shown to enhance digestive enzyme activity. Which strains have these properties? This needs to be further explained.

3. In the Materials and Methods section, proteolytic activity and anti-inflammatory methods, formula that has been used for the calculation implies that concentration equals intercept divided by absorbance. In colorimetric assays, absorbance usually increases with concentration. Your formula suggests an inverse relationship. You must verify this against your standard curve data. A standard curve typically follows the linear equation y = mx+c.

4. The statistical analysis reported for the analysis of the data is the calculation of the mean and standard deviation. This is not enough, and since different groups are being compared, statistical analysis such as One-Way ANOVA and Post-hoc Test need to be used. To confirm, for instance, strain A is superior to strain B, without the analysis of variance and a p-value, this is not acceptable and it requires statistical support.

5. In the results section, figure captions need to have a detailed explanation of what the figure is showing, including what each axis represents.

6. In the results section, the antagonistic activity section, PIB12RB strain, has been written twice.

7. In the results, proteolytic activity, why was the value of 0.365 selected? Also, in the same section, it has been reported that your consortium is less effective than the single strain of PIG5CI, with the value of 0.615 mg/mL. In the introduction and discussion, it has been claimed that the MSPC offers "synergistic effects." Scientifically, synergy implies that the combination is greater than the sum (or at least the best) of its parts. Here, your consortium is less effective than the single-strain PIG5CI. This needs to be further explained. In addition, mg/mL is not a common unit for the proteolytic activity as the activity products (tyrosine) have been measured and the no actual physical mass of the enzyme. The common unit is Units/mL, and this needs to be reviewed.

8. Table 1 and Figure 2A, Table 2 and Figure 3, present the same data, except for the MSPC value. This is redundant, and one should be selected to report the data.

9. In Figure 4, the data points jump from 7 hours to 24 hours. The actual growth curve has been missed and therefore, growth kinetics cannot be compared and discussed.

10. In the discussion section, the reported activity is 200 times lower than the literature you cited. Yet, it has been concluded: "Our results align with previous findings." This requires clarification and revision.

**Do you want your identity to be public for this peer review?** For information about this choice, including consent withdrawal, please see our Privacy Policy

Reviewer #1: No

Reviewer #2: No

---

## [Author Response · Author response to Decision Letter 1]

29 Jan 2026

Major Comments

1. Given that the study is conducted entirely in vitro, could the authors clarify whether such assays are sufficient to support the use of the term “potential therapeutic approach”?

Response to reviewer: We thank the reviewer for this important point. We agree that in vitro assays alone cannot establish therapeutic efficacy. In the revised manuscript, we have clarified that the term “potential therapeutic approach” is used to indicate preliminary functional evidence rather than clinical or in vivo validation. We have revised the Abstract, Discussion, and Conclusion to explicitly state that the findings are based on in vitro assays and that in vivo and clinical studies are required to confirm therapeutic applicability.

2. The authors include the sentence: “This study focuses on formulating and evaluating a multi-strain probiotic consortium (MSPC) as a potential therapeutic for PEM.” However, this statement only describes what was done and does not clearly express a specific research question or hypothesis. The authors should add a more precise objective or hypothesis.

Response to Reviewer: We appreciate this suggestion. The objective of the study has now been revised to clearly articulate the research question and rationale. A concise and specific objective statement has been added to the Introduction.

3. In the Statistical Analysis section, the authors state that the experiments “…were performed in triplicate”, but it is unclear whether this refers to technical replicates or biological replicates. This information is helpful for the reproducibility of the findings.

Response to Reviewer: We thank the reviewer for highlighting this point. The manuscript has been revised to clarify that all experiments were conducted using technical triplicates, unless otherwise specified.

4. Figures 2–5 contain multiple critical issues. Figure 2B displays negative enzyme concentrations, which are not physically meaningful. The error bars are unclear. The figures use inconsistent font sizes; the authors should adjust the figures so that all labeling uses a consistent font size. The horizontal grid lines should be removed unless they serve a clear purpose.

Response to Reviewer: We appreciate the detailed feedback. All figures have been carefully revised. Negative values have been corrected, error bars are now clearly defined in the figure legends, font sizes have been standardized across all figures, and unnecessary horizontal grid lines have been removed.

5. The results are reported with three decimal places throughout the figures and text. Could the authors clarify whether this level of precision reflects the actual accuracy of the assays and instruments used? (e.g, 0.528±0.03)

Response to Reviewer: We agree with the reviewer that excessive decimal precision may imply an unrealistic level of accuracy. The manuscript and figures have been revised to limit numerical values to two decimal places, consistent with assay sensitivity.

6. The manuscript includes a brief limitation statement (“Further research is needed…”, “Future clinical trials are warranted…”). Still, this language is quite general and does not specifically address the major methodological constraints of the study.

Response to Reviewer: We thank the reviewer for this constructive and valuable suggestion. We agree that the limitations should be more clearly articulated. The manuscript has been revised to explicitly state that the present work represents the first phase of a multi-stage research framework. This initial phase focuses on in vitro functional screening and consortium formulation, while subsequent phases are planned to include in vivo animal studies and metagenomic analyses to validate host–microbiome interactions and functional outcomes. These clarifications have been incorporated into the revised manuscript.

7. The authors may consider strengthening the Discussion by providing a deeper interpretation of the findings rather than primarily restating the results.

Response to Reviewer: We thank the reviewer for this insightful comment. We agree that the original Discussion was primarily descriptive. In the revised manuscript, the Discussion has been substantially rewritten to emphasize interpretation of the findings by explaining their biological relevance to protein-energy malnutrition, comparing the results with previously published studies, and highlighting how the observed functional properties of the consortium may contribute to nutritional support in PEM rather than restating numerical outcomes.

8. The conclusion would benefit from revision, because the current conclusion mainly restates earlier content; it does not provide a meaningful synthesis or deeper reflection on the implications and constraints of the study.

Response to Reviewer: The Conclusion has been revised to provide a concise synthesis of key findings, their implications, and the constraints of the study, rather than reiterating experimental details.

9. The manuscript provides some background on probiotics and PEM; however, I still find the rationale for selecting a Bacillus-based consortium somewhat limited. Would the authors consider expanding the justification for why these specific strains are suitable candidates for PEM intervention?

Response to Reviewer: We thank the reviewer for this valuable suggestion. The Introduction and Discussion have been expanded to better justify the selection of Bacillus species, emphasizing their spore-forming ability, environmental resilience, enzymatic activity, and stability under storage and gastrointestinal conditions.

Minor comments

1. Since the probiotic consortium consists of three Bacillus species, it may be helpful to specify "Bacillus-based multi-strain probiotic consortium" to improve clarity.

Response to Reviewer: We thank the reviewer for this helpful suggestion. To improve clarity and precision, we have revised the manuscript to consistently refer to the formulation as a “Bacillus-based multi-strain probiotic consortium” at appropriate locations.

2. “Formulation and functional evaluation of multi-strain probiotic consortium” should be “a multi-strain probiotic consortium.” This may cause readability issues.

Response to Reviewer: We thank the reviewer for pointing out this grammatical issue. The title and corresponding text have been revised to improve readability and grammatical accuracy.

3. Introducing the rationale for the MSPC earlier in the introduction could improve clarity. Bringing this information forward would create a smoother transition from the general discussion of malnutrition and probiotics to the specific focus of the study.

Response to Reviewer: We appreciate this suggestion. The Introduction has been revised to introduce the rationale for the multi-strain probiotic consortium earlier, thereby improving the logical flow from general background to study-specific focus.

4. In the introduction section, please replace the “&” symbol in the phrase “small peptides & amino acids” with “and”.

Response to Reviewer: We thank the reviewer for this observation. The ampersand symbol has been replaced with “and” to maintain formal scientific writing style.

5. In the introduction, the sentence beginning with “Over the past decade, MSPC have shown…” uses the abbreviation MSPC as a plural noun. Using “MSPC has shown…” would be more grammatically appropriate.

Response to Reviewer: We appreciate the reviewer’s attention to grammatical accuracy. The sentence has been revised to ensure correct singular usage of the abbreviation.

6. Space Inconsistencies in Percentages. In the introduction, “According to a report by the World Health Organization (WHO)…” contains spacing issues.

Response to Reviewer: We thank the reviewer for noting this formatting issue. Spacing inconsistencies in percentage values have been corrected throughout the manuscript to ensure uniform formatting.

7. A careful double-check of the references would help ensure consistent formatting throughout the manuscript.

Response to Reviewer: We appreciate this comment. The reference list has been carefully reviewed and reformatted to ensure consistency in journal abbreviations, author formatting, and DOI presentation, in accordance with journal guidelines.

Comments of portal

1. In the introduction section, when talking about malnutrition in India, supporting statistics needs to be added. Without any data, the claim in the study feels vague.

Response to Reviewer: We thank the reviewer for this important observation. We agree that quantitative data are essential to strengthen the context of malnutrition in India. Accordingly, the Introduction has been revised to include recent and authoritative national and global statistics highlighting the prevalence and burden of malnutrition in India.

2. In the introduction section, in the 2nd paragraph, it has been reported that several probiotic strains have been shown to enhance digestive enzyme activity. Which strains have these properties? This needs to be further explained.

Response to Reviewer: We appreciate the reviewer’s request for clarification. The Introduction has been revised to explicitly identify probiotic genera and representative strains reported to enhance digestive enzyme activity, thereby improving clarity and scientific grounding.

3. In the Materials and Methods section, proteolytic activity and anti-inflammatory methods, formula that has been used for the calculation implies that concentration equals intercept divided by absorbance. In colorimetric assays, absorbance usually increases with concentration. Your formula suggests an inverse relationship. You must verify this against your standard curve data. A standard curve typically follows the linear equation y = mx+c.

Response to Reviewer: We thank the reviewer for carefully identifying this methodological inconsistency. We have re-verified the standard curve data and corrected the calculation method to reflect the appropriate linear relationship between absorbance and concentration based on the equation y = mx + c.

4. The statistical analysis reported for the analysis of the data is the calculation of the mean and standard deviation. This is not enough, and since different groups are being compared, statistical analysis such as One-Way ANOVA and Post-hoc Test need to be used. To confirm, for instance, strain A is superior to strain B, without the analysis of variance and a p-value, this is not acceptable and it requires statistical support.

Response to Reviewer: We thank the reviewer for this important comment. The primary objective of the present study was functional screening and characterization of individual strains and the formulated consortium, rather than hypothesis-driven comparison between experimental treatment groups. Each strain and the MSPC were evaluated independently under identical assay conditions, and results were therefore summarized using descriptive statistics (mean ± standard deviation). For this, we have reframed our statements in result and discussion. We have removed words like significantly higher, statistically superior, and significant difference.

Since no experimental grouping or treatment-based comparison was performed, inferential statistical tests such as one-way ANOVA were not applied. This approach is consistent with exploratory in vitro screening studies aimed at identifying functional potential rather than establishing statistical superiority between strains. We have clarified this rationale in the Statistical Analysis section of the manuscript.

5. In the results section, figure captions need to have a detailed explanation of what the figure is showing, including what each axis represents.

Response to Reviewer: We thank the reviewer for this suggestion. All figure captions have been revised to clearly describe the experimental context, axes, units, and comparisons shown. Also, as per comment no. 8 below, the respective figure has been removed, and the table has been kept.

6. In the results section, the antagonistic activity section, PIB12RB strain, has been written twice.

Response to Reviewer: We thank the reviewer for pointing out this oversight. The duplication of the PIB12RB strain in the antagonistic activity section has been corrected.

7. In the results, proteolytic activity, why was the value of 0.365 selected? Also, in the same section, it has been reported that your consortium is less effective than the single strain of PIG5CI, with the value of 0.615 mg/mL. In the introduction and discussion, it has been claimed that the MSPC offers "synergistic effects." Scientifically, synergy implies that the combination is greater than the sum (or at least the best) of its parts. Here, your consortium is less effective than the single-strain PIG5CI. This needs to be further explained. In addition, mg/mL is not a common unit for the proteolytic activity as the activity products (tyrosine) have been measured and the no actual physical mass of the enzyme. The common unit is Units/mL, and this needs to be reviewed.

Response to Reviewer: We thank the reviewer for this insightful comment. The selection of the proteolytic activity threshold and the interpretation of synergistic effects have been clarified, and the unit of proteolytic activity has been corrected throughout the manuscript.

The value of PIG3IR (0.365 Units/mL) was not selected arbitrarily. This strain was prioritized because, in addition to exhibiting moderate proteolytic activity, it demonstrated superior anti-inflammatory activity compared to other isolates. Given the multifactorial nature of protein-energy malnutrition, strain selection for consortium development was based on overall functional performance, rather than maximal activity in a single assay.

Furthermore, the interpretation of “synergy” has been revised to reflect functional complementarity rather than numerical enhancement of a single parameter. Although the MSPC exhibited lower proteolytic activity than the single strain PIG5CI, the consortium was designed to integrate complementary enzymatic, antioxidant, and anti-inflammatory properties, along with improved strain compatibility, to achieve broader functional relevance.

Finally, proteolytic activity units have been corrected from mg/mL to Units/mL.

8. Table 1 and Figure 2A, Table 2 and Figure 3, present the same data, except for the MSPC value. This is redundant, and one should be selected to report the data.

Response to Reviewer: We agree with the reviewer that presenting identical datasets in both tabular and graphical formats is redundant. The data presentation has been streamlined by retaining only one format.

9. In Figure 4, the data points jump from 7 hours to 24 hours. The actual growth curve has been missed and therefore, growth kinetics cannot be compared and discussed.

Response to Reviewer: We thank the reviewer for raising this point. The growth curve analysis was primarily designed to monitor early growth behavior up to 7 h. However, additional measurements were performed at 24 h and 48 h to confirm entry into and maintenance of the stationary phase. Intermediate time points between 7 h and 24 h were therefore not included, which limited detailed kinetic comparisons during the exponential phase. This limitation has now been explicitly acknowledged in the revised Discussion.

10. In the discussion section, the reported activity is 200 times lower than the literature you cited. Yet, it has been concluded: "Our results align with previous findings." This requires clarification and revision.

Response to Reviewer: We thank the reviewer for highlighting this important discrepancy. The Discussion has been revised to avoid overstating alignment with previous studies and to contextualize the lower magnitude of activity in terms of strain specificity and assay conditions.

---

## [Decision Letter · Decision Letter 1]

4 Mar 2026

Dear Dr. Sarkar,

Thank you for submitting your manuscript to PLOS ONE. After careful consideration, we feel that it has merit but does not fully meet PLOS ONE’s publication criteria as it currently stands. Therefore, we invite you to submit a revised version of the manuscript that addresses the points raised during the review process.

We look forward to receiving your revised manuscript.

Kind regards,

Babak Pakbin

Academic Editor

PLOS One

**Journal Requirements:**

Reviewers' comments:

Reviewer's Responses to Questions

**Comments to the Author**

Reviewer #1: All comments have been addressed

Reviewer #2: All comments have been addressed

2. Is the manuscript technically sound, and do the data support the conclusions?

Reviewer #1: Partly

Reviewer #2: (No Response)

3. Has the statistical analysis been performed appropriately and rigorously?

Reviewer #1: I Don't Know

Reviewer #2: (No Response)

4. Have the authors made all data underlying the findings in their manuscript fully available?

Reviewer #1: Yes

Reviewer #2: (No Response)

5. Is the manuscript presented in an intelligible fashion and written in standard English?

Reviewer #1: Yes

Reviewer #2: (No Response)

**Reviewer #1:**  The authors have adequately addressed the comments raised in the previous round of review. The revisions have improved the clarity and scientific presentation of the manuscript. However, further refinement in the interpretation of the findings, overall flow of writing, and clearer presentation of error bars in the figures would strengthen the manuscript.

**Reviewer #2:**  (No Response)

**Do you want your identity to be public for this peer review?** For information about this choice, including consent withdrawal, please see our Privacy Policy

Reviewer #1: No

Reviewer #2: No

---

## [Author Response · Author response to Decision Letter 2]

9 Mar 2026

Response to Reviewers

Manuscript ID: PONE-D-25-60367R1

Title: Formulation and in-vitro functional evaluation of a Bacillus-based multi-strain probiotic consortium relevant to protein-energy malnutrition

Dear Editor and Reviewers,

We sincerely thank the Academic Editor and the reviewers for their careful evaluation of our manuscript and for the constructive comments provided during the review process. We appreciate the positive feedback and are grateful for the opportunity to revise our manuscript.

We are pleased to note that the reviewers acknowledged that the comments from the previous round have been adequately addressed and that the revisions improved the clarity and scientific presentation of the manuscript. Following the reviewer’s suggestions, we have further revised the manuscript to refine the interpretation of the findings, improve the overall flow of the text, and enhance the clarity of error bar presentation in the figures.

All changes made in the revised manuscript are clearly highlighted in the marked-up version submitted as “Revised Manuscript with Track Changes.” An unmarked, clean version of the manuscript has also been provided as requested.

Below, we provide our point-by-point response to the reviewer’s comment.

Reviewer - 1

Comment: The authors have adequately addressed the comments raised in the previous round of review. The revisions have improved the clarity and scientific presentation of the manuscript. However, further refinement in the interpretation of the findings, overall flow of writing, and clearer presentation of error bars in the figures would strengthen the manuscript.

Response:

We sincerely thank the reviewer for the positive assessment of our revised manuscript and for the valuable suggestions to further improve its quality.

Refinement of the interpretation of findings

The discussion section has been carefully revised to provide a clearer and more precise interpretation of the experimental findings. Additional explanations have been incorporated to better relate the results to the functional characteristics of the Bacillus-based probiotic consortium and its potential relevance to protein-energy malnutrition.

Improvement in the overall flow of writing

The manuscript text has been edited throughout to improve readability and logical flow. Sentences and paragraph transitions in the Results and Discussion sections have been refined to ensure clearer progression of ideas.

Clearer presentation of error bars in figures

The figures have been revised to improve the visibility and clarity of the error bars. The figure legends have also been updated to explicitly state the statistical parameters represented by the error bars (mean ± standard deviation).

We believe these revisions have further improved the clarity, presentation, and scientific interpretation of the manuscript.

Once again, we sincerely thank the editor and reviewer for their constructive feedback and for considering our revised manuscript. We hope that the changes made satisfactorily address the reviewer’s concerns and that the manuscript is now suitable for publication.

Sincerely,

Priya Mori

(On behalf of all authors)

---

## [Editor Report · Decision Letter 2]

11 Mar 2026

Formulation and in-vitro functional evaluation of a Bacillus-based multi-strain probiotic consortium relevant to protein-energy malnutrition

PONE-D-25-60367R2

Dear Dr. Sarkar,

We’re pleased to inform you that your manuscript has been judged scientifically suitable for publication and will be formally accepted for publication once it meets all outstanding technical requirements.

Kind regards,

Babak Pakbin

Academic Editor

PLOS One
---

## [Editor Report · Acceptance letter]

PONE-D-25-60367R2

PLOS One

Dear Dr. Sarkar,

I'm pleased to inform you that your manuscript has been deemed suitable for publication in PLOS One. Congratulations! Your manuscript is now being handed over to our production team.

Kind regards,

on behalf of

Dr. Babak Pakbin

Academic Editor

PLOS One